# On the consistency theory of high dimensional variable screening

**Xiangyu Wang**
Dept. of Statistical Science
Duke University, USA
xw56@stat.duke.edu

**Chenlei Leng**
Dept. of Statistics
University of Warwick, UK
C.Leng@warwick.ac.uk

**David B. Dunson**
Dept. of Statistical Science
Duke University, USA
dunson@stat.duke.edu

## Abstract

Variable screening is a fast dimension reduction technique for assisting high dimensional feature selection. As a preselection method, it selects a moderate size subset of candidate variables for further refining via feature selection to produce the final model. The performance of variable screening depends on both computational efficiency and the ability to dramatically reduce the number of variables without discarding the important ones. When the data dimension $p$ is substantially larger than the sample size $n$, variable screening becomes crucial as 1) Faster feature selection algorithms are needed; 2) Conditions guaranteeing selection consistency might fail to hold. This article studies a class of linear screening methods and establishes consistency theory for this special class. In particular, we prove the restricted diagonally dominant (RDD) condition is a necessary and sufficient condition for strong screening consistency. As concrete examples, we show two screening methods $SIS$ and $HOLP$ are both strong screening consistent (subject to additional constraints) with large probability if $n > O((\rho s + \sigma/\tau)^2 \log p)$ under random designs. In addition, we relate the RDD condition to the irrepresentable condition, and highlight limitations of $SIS$.

## 1 Introduction

The rapidly growing data dimension has brought new challenges to statistical variable selection, a crucial technique for identifying important variables to facilitate interpretation and improve prediction accuracy. Recent decades have witnessed an explosion of research in variable selection and related fields such as compressed sensing [1, 2], with a core focus on regularized methods [3–7]. Regularized methods can consistently recover the support of coefficients, i.e., the non-zero signals, via optimizing regularized loss functions under certain conditions [8–10]. However, in the big data era when $p$ far exceeds $n$, such regularized methods might fail due to two reasons. First, the conditions that guarantee variable selection consistency for convex regularized methods such as *lasso* might fail to hold when $p >> n$; Second, the computational expense of both convex and non-convex regularized methods increases dramatically with large $p$.

Bearing these concerns in mind, [11] propose the concept of "variable screening", a fast technique that reduces data dimensionality from $p$ to a size comparable to $n$, with all predictors having non-zero coefficients preserved. They propose a marginal correlation based fast screening technique "Sure Independence Screening" ($SIS$) that can preserve signals with large probability. However, this method relies on a strong assumption that the marginal correlations between the response and the important predictors are high [11], which is easily violated in the practice. [12] extends the marginal correlation to the Spearman's rank correlation, which is shown to gain certain robustness but is still limited by the same strong assumption. [13] and [14] take a different approach to attack the screening problem. They both adopt variants of a forward selection type algorithm that includes one variable at a time for constructing a candidate variable set for further refining. These methods

eliminate the strong marginal assumption in [11] and have been shown to achieve better empirical performance. However, such improvement is limited by the extra computational burden caused by their iterative framework, which is reported to be high when $p$ is large [15]. To ameliorate concerns in both screening performance and computational efficiency, [15] develop a new type of screening method termed "High-dimensional ordinary least-square projection" ($HOLP$). This new screener relaxes the strong marginal assumption required by $SIS$ and can be computed efficiently (complexity is $O(n^2 p)$), thus scalable to ultra-high dimensionality.

This article focuses on linear models for tractability. As computation is one vital concern for designing a good screening method, we primarily focus on a class of linear screeners that can be efficiently computed, and study their theoretical properties. The main contributions of this article lie in three aspects.

1. We define the notion of strong screening consistency to provide a unified framework for analyzing screening methods. In particular, we show a necessary and sufficient condition for a screening method to be strong screening consistent is that the screening matrix is restricted diagonally dominant (RDD). This condition gives insights into the design of screening matrices, while providing a framework to assess the effectiveness of screening methods.

2. We relate RDD to other existing conditions. The irrepresentable condition (IC) [8] is necessary and sufficient for sign consistency of lasso [3]. In contrast to IC that is specific to the design matrix, RDD involves another ancillary matrix that can be chosen arbitrarily. Such flexibility allows RDD to hold even when IC fails if the ancillary matrix is carefully chosen (as in $HOLP$). When the ancillary matrix is chosen as the design matrix, certain equivalence is shown between RDD and IC, revealing the difficulty for $SIS$ to achieve screening consistency. We also comment on the relationship between RDD and the restricted eigenvalue condition (REC) [6] which is commonly seen in the high dimensional literature. We illustrate via a simple example that RDD might not be necessarily stronger than REC.

3. We study the behavior of $SIS$ and $HOLP$ under random designs, and prove that a sample size of $n = O\big((\rho s + \sigma/\tau)^2 \log p\big)$ is sufficient for $SIS$ and $HOLP$ to be screening consistent, where $s$ is the sparsity, $\rho$ measures the diversity of signals and $\tau/\sigma$ evaluates the signal-to-noise ratio. This is to be compared to the sign consistency results in [9] where the design matrix is fixed and assumed to follow the IC.

The article is organized as follows. In Section 1, we set up the basic problem and describe the framework of variable screening. In Section 2, we provide a deterministic necessary and sufficient condition for consistent screening. Its relationship with the irrepresentable condition is discussed in Section 3. In Section 4, we prove the consistency of $SIS$ and $HOLP$ under random designs by showing the RDD condition is satisfied with large probability, although the requirement on $SIS$ is much more restictive.

## 2 Linear screening

Consider the usual linear regression

$$Y = X\beta + \epsilon,$$

where $Y$ is the $n \times 1$ response vector, $X$ is the $n \times p$ design matrix and $\epsilon$ is the noise. The regression task is to learn the coefficient vector $\beta$. In the high dimensional setting where $p >> n$, a sparsity assumption is often imposed on $\beta$ so that only a small portion of the coordinates are non-zero. Such an assumption splits the task of learning $\beta$ into two phases. The first is to recover the support of $\beta$, i.e., the location of non-zero coefficients; The second is to estimate the value of these non-zero signals. This article mainly focuses on the first phase.

As pointed out in the introduction, when the dimensionality is too high, using regularization methods methods raises concerns both computationally and theoretically. To reduce the dimensionality, [11] suggest a variable screening framework by finding a submodel

$$\mathcal{M}_d = \{i \ : \ |\hat{\beta}_i| \text{ is among the largest d coordinates of } |\hat{\beta}|\} \quad \text{or} \quad \mathcal{M}_\gamma = \{i \ : \ |\hat{\beta}_i| > \gamma\}.$$

Let $Q = \{1, 2, \cdots, p\}$ and define $S$ as the true model with $s = |S|$ being its cardinarlity. The hope is that the submodel size $|\mathcal{M}_d|$ or $|\mathcal{M}_\gamma|$ will be smaller or comparable to $n$, while $S \subseteq \mathcal{M}_d$ or $S \subseteq \mathcal{M}_\gamma$. To achieve this goal two steps are usually involved in the screening analysis. The first is to show there exists some $\gamma$ such that $\min_{i \in S} |\hat{\beta}_i| > \gamma$ and the second step is to bound the size of $|\mathcal{M}_\gamma|$ such that $|\mathcal{M}_\gamma| = O(n)$. To unify these steps for a more comprehensive theoretical framework, we put forward a slightly stronger definition of screening consistency in this article.

**Definition 2.1.** *(Strong screening consistency) An estimator $\hat{\beta}$ (of $\beta$) is strong screening consistent if it satisfies that*

$$\min_{i \in S} |\hat{\beta}_i| > \max_{i \notin S} |\hat{\beta}_i| \tag{1}$$

*and*

$$sign(\hat{\beta}_i) = sign(\beta_i), \quad \forall i \in S. \tag{2}$$

**Remark 2.1.** *This definition does not differ much from the usual screening property studied in the literature, which requires $\min_{i \in S} |\hat{\beta}_i| > \max_{i \notin S}^{(n-s)} |\hat{\beta}_i|$, where $\max^{(k)}$ denotes the $k^{th}$ largest item.*

The key of strong screening consistency is the property (1) that requires the estimator to preserve consistent ordering of the zero and non-zero coefficients. It is weaker than variable selection consistency in [8]. The requirement in (2) can be seen as a relaxation of the sign consistency defined in [8], as no requirement for $\hat{\beta}_i, i \notin S$ is needed. As shown later, such relaxation tremendously reduces the restriction on the design matrix, and allows screening methods to work for a broader choice of $X$.

The focus of this article is to study the theoretical properties of a special class of screeners that take the linear form as

$$\hat{\beta} = AY$$

for some $p \times n$ ancillary matrix $A$. Examples include sure independence screening ($SIS$) where $A = X^T/n$ and high-dimensional ordinary least-square projection ($HOLP$) where $A = X^T(XX^T)^{-1}$. We choose to study the class of linear estimators because linear screening is computationally efficient and theoretically tractable. We note that the usual ordinary least-squares estimator is also a special case of linear estimators although it is not well defined for $p > n$.

## 3 Deterministic guarantees

In this section, we derive the necessary and sufficient condition that guarantees $\hat{\beta} = AY$ to be strong screening consistent. The design matrix $X$ and the error $\epsilon$ are treated as fixed in this section and we will investigate random designs later. We consider the set of sparse coefficient vectors defined by

$$\mathcal{B}(s, \rho) = \left\{ \beta \in \mathcal{R}^p : |supp(\beta)| \leq s, \quad \frac{\max_{i \in supp(\beta)} |\beta_i|}{\min_{i \in supp(\beta)} |\beta_i|} \leq \rho \right\}.$$

The set $\mathcal{B}(s, \rho)$ contains vectors having at most $s$ non-zero coordinates with the ratio of the largest and smallest coordinate bounded by $\rho$. Before proceeding to the main result of this section, we introduce some terminology that helps to establish the theory.

**Definition 3.1.** *(restricted diagonally dominant matrix) A $p \times p$ symmetric matrix $\Phi$ is restricted diagonally dominant with sparsity $s$ if for any $I \subseteq Q$, $|I| \leq s - 1$ and $i \in Q \setminus I$*

$$\Phi_{ii} > C_0 \max \left\{ \sum_{j \in I} |\Phi_{ij} + \Phi_{kj}|, \sum_{j \in I} |\Phi_{ij} - \Phi_{kj}| \right\} + |\Phi_{ik}| \quad \forall k \neq i, \ k \in Q \setminus I,$$

*where $C_0 \geq 1$ is a constant.*

Notice this definition implies that for $i \in Q \setminus I$

$$\Phi_{ii} \geq C_0 \left( \sum_{j \in I} |\Phi_{ij} + \Phi_{kj}| + \sum_{j \in I} |\Phi_{ij} - \Phi_{kj}| \right)/2 \geq C_0 \sum_{j \in I} |\Phi_{ij}|, \tag{3}$$

which is related to the usual diagonally dominant matrix. The restricted diagonally dominant matrix provides a necessary and sufficient condition for any linear estimators $\hat{\beta} = AY$ to be strong screening consistent. More precisely, we have the following result.

**Theorem 1.** *For the noiseless case where $\epsilon = 0$, a linear estimator $\hat{\beta} = AY$ is strong screening consistent for every $\beta \in \mathcal{B}(s, \rho)$, if and only if the screening matrix $\Phi = AX$ is restricted diagonally dominant with sparsity $s$ and $C_0 \geq \rho$.*

*Proof.* Assume $\Phi$ is restricted diagonally dominant with sparsity $s$ and $C_0 \geq \rho$. Recall $\hat{\beta} = \Phi\beta$. Suppose $S$ is the index set of non-zero predictors. For any $i \in S, k \notin S$, if we let $I = S \setminus \{i\}$, then we have

$$|\hat{\beta}_i| = |\beta_i|\left(\Phi_{ii} + \sum_{j\in I}\frac{\beta_j}{\beta_i}\Phi_{ij}\right) = |\beta_i|\left\{\Phi_{ii} + \sum_{j\in I}\frac{\beta_j}{\beta_i}(\Phi_{ij} + \Phi_{kj}) + \Phi_{ki} - \sum_{j\in I}\frac{\beta_j}{\beta_i}\Phi_{kj} - \Phi_{ki}\right\}$$

$$> -|\beta_i|\left(\sum_{j\in I}\frac{\beta_j}{\beta_i}\Phi_{kj} + \Phi_{ki}\right) = -\frac{|\beta_i|}{\beta_i}\left(\sum_{j\in I}\beta_j\Phi_{kj} + \beta_i\Phi_{ki}\right) = -sign(\beta_i)\cdot\hat{\beta}_k,$$

and

$$|\hat{\beta}_i| = |\beta_i|\left(\Phi_{ii} + \sum_{j\in I}\frac{\beta_j}{\beta_i}\Phi_{ij}\right) = |\beta_i|\left\{\Phi_{ii} + \sum_{j\in I}\frac{\beta_j}{\beta_i}(\Phi_{ij} - \Phi_{kj}) - \Phi_{ki} + \sum_{j\in I}\frac{\beta_j}{\beta_i}\Phi_{kj} + \Phi_{ki}\right\}$$

$$> |\beta_i|\left(\sum_{j\in I}\frac{\beta_j}{\beta_i}\Phi_{kj} + \Phi_{ki}\right) = sign(\beta_i)\cdot\hat{\beta}_k.$$

Therefore, whatever value $sign(\beta_i)$ is, it always holds that $|\hat{\beta}_i| > |\hat{\beta}_k|$ and thus $\min_{i\in S}|\hat{\beta}_i| > \max_{k\notin S}|\hat{\beta}_k|$.

To prove the sign consistency for non-zero coefficients, we notice that for $i \in S$,

$$\hat{\beta}_i\beta_i = \Phi_{ii}\beta_i^2 + \sum_{j\in I}\Phi_{ij}\beta_j\beta_i = \beta_i^2\left(\Phi_{ii} + \sum_{j\in I}\frac{\beta_j}{\beta_i}\Phi_{ij}\right) > 0.$$

The proof of necessity is left to the supplementary materials.

$\square$

The noiseless case is a good starting point to analyze $\hat{\beta}$. Intuitively, in order to preserve the correct order of the coefficients in $\hat{\beta} = AX\beta$, one needs $AX$ to be close to a diagonally dominant matrix, so that $\hat{\beta}_i, i \in \mathcal{M}_S$ will take advantage of the large diagonal terms in $AX$ to dominate $\hat{\beta}_i, i \notin \mathcal{M}_S$ that is just linear combinations of off-diagonal terms.

When noise is considered, the condition in Theorem 1 needs to be changed slightly to accommodate extra discrepancies. In addition, the smallest non-zero coefficient has to be lower bounded to ensure a certain level of signal-to-noise ratio. Thus, we augment our previous definition of $\mathcal{B}(s, \rho)$ to have a signal strength control

$$\mathcal{B}_\tau(s, \rho) = \{\beta \in \mathcal{B}(s, \rho)|\min_{i\in supp(\beta)}|\beta_i| \geq \tau\}.$$

Then we can obtain the following modified Theorem.

**Theorem 2.** *With noise, the linear estimator $\hat{\beta} = AY$ is strong screening consistent for every $\beta \in \mathcal{B}_\tau(s, \rho)$ if $\Phi = AX - 2\tau^{-1}\|A\epsilon\|_\infty I_p$ is restricted diagonally dominant with sparsity $s$ and $C_0 \geq \rho$.*

The proof of Theorem 2 is essentially the same as Theorem 1 and is thus left to the supplementary materials. The condition in Theorem 2 can be further tailored to a necessary and sufficient version with extra manipulation on the noise term. Nevertheless, this might not be useful in practice due to the randomness in noise. In addition, the current version of Theorem 2 is already tight in the sense that there exists some noise vector $\epsilon$ such that the condition in Theorem 2 is also necessary for strong screening consistency.

Theorems 1 and 2 establish ground rules for verifying consistency of a given screener and provide practical guidance for screening design. In Section 4, we consider some concrete examples of ancillary matrix $A$ and prove that conditions in Theorems 1 and 2 are satisfied by the corresponding screeners with large probability under random designs.

# 4 Relationship with other conditions

For some special cases such sure independence screening ("SIS"), the restricted diagonally dominant (RDD) condition is related to the strong irrepresentable condition (IC) proposed in [8]. Assume each column of $X$ is standardized to have mean zero. Letting $C = X^T X/n$ and $\beta$ be a given coefficient vector, the IC is expressed as

$$\|C_{S^c,S} C_{S,S}^{-1} \cdot sign(\beta_S)\|_\infty \leq 1 - \theta \tag{4}$$

for some $\theta > 0$, where $C_{A,B}$ represents the sub-matrix of $C$ with row indices in $A$ and column indices in $B$. The authors enumerate several scenarios of $C$ such that IC is satisfied. We verify some of these scenarios for screening matrix $\Phi$.

**Corollary 1.** *If $\Phi_{ii} = 1$, $\forall i$ and $|\Phi_{ij}| < c/(2s)$, $\forall i \neq j$ for some $0 \leq c < 1$ as defined in Corollary 1 and 2 in [8], then $\Phi$ is a restricted diagonally dominant matrix with sparsity $s$ and $C_0 \geq 1/c$.*

*If $|\Phi_{ij}| < r^{|i-j|}$, $\forall i, j$ for some $0 < r < 1$ as defined in Corollary 3 in [8], then $\Phi$ is a restricted diagonally dominant matrix with sparsity $s$ and $C_0 \geq (1-r)^2/(4r)$.*

A more explicit but nontrivial relationship between IC and RDD is illustrated below when $|S| = 2$.

**Theorem 3.** *Assume $\Phi_{ii} = 1$, $\forall i$ and $|\Phi_{ij}| < r$, $\forall i \neq j$. If $\Phi$ is restricted diagonally dominant with sparsity 2 and $C_0 \geq \rho$, then $\Phi$ satisfies*

$$\|\Phi_{S^c,S} \Phi_{S,S}^{-1} \cdot sign(\beta_S)\|_\infty \leq \frac{\rho^{-1}}{1-r}$$

*for all $\beta \in \mathcal{B}(2, \rho)$. On the other hand, if $\Phi$ satisfies the IC for all $\beta \in \mathcal{B}(2, \rho)$ for some $\theta$, then $\Phi$ is a restricted diagonally dominant matrix with sparsity 2 and*

$$C_0 \geq \frac{1}{1-\theta} \frac{1-r}{1+r}.$$

Theorem 3 demonstrates certain equivalence between IC and RDD. However, it does not mean that RDD is also a strong requirement. Notice that IC is directly imposed on the covariance matrix $X^T X/n$. This makes IC a strong assumption that is easily violated; for example, when the predictors are highly correlated. In contrast to IC, RDD is imposed on matrix $AX$ where there is flexibility in choosing $A$. Only when $A$ is chose to be $X/n$, RDD is equivalently strong as IC, as shown in next theorem. For other choices of $A$, such as $HOLP$ defined in next section, the estimator satisfies RDD even when predictors are highly correlated. Therefore, RDD is considered as weak requirement.

For "SIS", the screening matrix $\Phi = X^T X/n$ coincides with the covariance matrix, making RDD and IC effectively equivalent. The following theorem formalizes this.

**Theorem 4.** *Let $A = X^T/n$ and standardize columns of $X$ to have sample variance one. Assume $X$ satisfies the sparse Riesz condition [16], i.e,*

$$\min_{\pi \subseteq Q, \ |\pi| \leq s} \lambda_{min}(X_\pi^T X_\pi/n) \geq \mu,$$

*for some $\mu > 0$. Now if $AX$ is restricted diagonally dominant with sparsity $s + 1$ and $C_0 \geq \rho$ with $\rho > \sqrt{s}/\mu$, then $X$ satisfies the IC for any $\beta \in \mathcal{B}(s, \rho)$.*

*In other words, under the condition $\rho > \sqrt{s}/\mu$, the strong screening consistency of SIS for $\mathcal{B}(s + 1, \rho)$ implies the model selection consistency of lasso for $\mathcal{B}(s, \rho)$.*

Theorem 4 illustrates the difficulty of $SIS$. The necessary condition that guarantees good screening performance of $SIS$ also guarantees the model selection consistency of lasso. However, such a strong necessary condition does not mean that $SIS$ should be avoided in practice given its substantial advantages in terms of simplicity and computational efficiency. The strong screening consistency defined in this article is stronger than conditions commonly used in justifying screening procedures as in [11].

Another common assumption in the high dimensional literature is the restricted eigenvalue condition (REC). Compared to REC, RDD is not necessarily stronger due to its flexibility in choosing the ancillary matrix $A$. [17, 18] prove that the REC is satisfied when the design matrix is sub-Gaussian. However, REC might not be guaranteed when the row of $X$ follows heavy-tailed distribution. In contrast, as the example shown in next section and in [15], by choosing $A = X^T(XX^T)^{-1}$, the resulting estimator satisfies RDD even when the rows of $X$ follow heavy-tailed distributions.

# 5 Screening under random designs

In this section, we consider linear screening under random designs when $X$ and $\epsilon$ are Gaussian. The theory developed in this section can be easily extended to a broader family of distributions, for example, where $\epsilon$ follows a sub-Gaussian distribution [19] and $X$ follows an elliptical distribution [11, 15]. We focus on the Gaussian case for conciseness. Let $\epsilon \sim N(0, \sigma^2)$ and $X \sim N(0, \Sigma)$. We prove the screening consistency of $SIS$ and $HOLP$ by verifying the condition in Theorem 2. Recall the ancillary matrices for $SIS$ and $HOLP$ are defined respectively as

$$A_{SIS} = X/n, \qquad A_{HOLP} = X^T(XX^T)^{-1}.$$

For simplicity, we assume $\Sigma_{ii} = 1$ for $i = 1, 2, \cdots, p$. To verify the RDD condition, it is essential to quantify the magnitude of the entries of $AX$ and $A\epsilon$.

**Lemma 1.** *Let $\Phi = A_{SIS}X$, then for any $t > 0$ and $i \neq j \in Q$, we have*

$$P\left(|\Phi_{ii} - \Sigma_{ii}| \geq t\right) \leq 2\exp\left\{-\min\left(\frac{t^2 n}{8e^2 K}, \frac{tn}{2eK}\right)\right\},$$

*and*

$$P\left(|\Phi_{ij} - \Sigma_{ij}| \geq t\right) \leq 6\exp\left\{-\min\left(\frac{t^2 n}{72e^2 K}, \frac{tn}{6eK}\right)\right\},$$

*where $K = \|\mathcal{X}^2(1) - 1\|_{\psi_1}$ is a constant, $\mathcal{X}^2(1)$ is a chi-square random variable with one degree of freedom and the norm $\|\cdot\|_{\psi_1}$ is defined in [19].*

Lemma 1 states that the screening matrix $\Phi = A_{SIS}X$ for SIS will eventually converge to the covariance matrix $\Sigma$ in $l_\infty$ when $n$ tends to infinity and $\log p = o(n)$. Thus, the screening performance of SIS strongly relies on the structure of $\Sigma$. In particular, the (asymptotically) necessary and sufficient condition for $SIS$ being strong screening consistent is $\Sigma$ satisfying the RDD condition. For the noise term, we have the following lemma.

**Lemma 2.** *Let $\eta = A_{SIS}\epsilon$. For any $t > 0$ and $i \in Q$, we have*

$$P(|\eta_i| \geq \sigma t) \leq 6\exp\left\{-\min\left(\frac{t^2 n}{72e^2 K}, \frac{tn}{6eK}\right)\right\},$$

*where $K$ is defined the same as in Lemma 1.*

The proof of Lemma 2 is essentially the same as the proof of off-diagonal terms in Lemma 1 and is thus omitted. As indicated before, the necessary and sufficient condition for $SIS$ to be strong screening consistent is that $\Sigma$ follows RDD. As RDD is usually hard to verify, we consider a stronger sufficient condition inspired by Corollary 1.

**Theorem 5.** *Let $r = \max_{i \neq j} |\Sigma_{ij}|$. If $r < \frac{1}{2\rho s}$, then for any $\delta > 0$, if the sample size satisfies*

$$n > 144K\left(\frac{1 + 2\rho s + 2\sigma/\tau}{1 - 2\rho sr}\right)^2 \log(3p/\delta), \tag{5}$$

*where $K$ is defined in Lemma 1, then with probability at least $1 - \delta$, $\Phi = A_{SIS}X - 2\tau^{-1}\|A_{SIS}\epsilon\|_\infty I_p$ is restricted diagonally dominant with sparsity $s$ and $C_0 \geq \rho$. In other words, SIS is screening consistent for any $\beta \in \mathcal{B}_\tau(s, \rho)$.*

*Proof.* Taking union bound on the results from Lemma 1 and 2, we have for any $t > 0$ and $p > 2$,

$$P\left(\min_{i \in Q} \Phi_{ii} \leq 1 - t \text{ or } \max_{i \neq j} |\Phi_{ij}| \geq r + t \text{ or } \|\eta\|_\infty \geq \sigma t\right) \leq 7p^2 \exp\left\{-\frac{n}{K}\min\left(\frac{t^2}{72e^2}, \frac{t}{6e}\right)\right\}.$$

In other words, for any $\delta > 0$, when $n \geq K\log(7p^2/\delta)$, with probability at least $1 - \delta$, we have

$$\min_{i \in Q} \Phi_{ii} \geq 1 - 6\sqrt{2}e\sqrt{\frac{K\log(7p^2/\delta)}{n}}, \quad \max_{i \neq j} |\Phi_{ij}| \leq r + 6\sqrt{2}e\sqrt{\frac{K\log(7p^2/\delta)}{n}},$$

$$\max_{i \in Q} |\eta_i| \le 6\sqrt{2}e\sigma\sqrt{\frac{K\log(7p^2/\delta)}{n}}.$$

A sufficient condition for $\Phi$ to be restricted diagonally dominant is that

$$\min_i \Phi_{ii} > 2\rho s \max_{i \ne j} |\Phi_{ij}| + 2\tau^{-1}\max_i |\eta_i|.$$

Plugging in the values we have

$$1 - 6\sqrt{2}e\sqrt{\frac{K\log(7p^2/\delta)}{n}} > 2\rho s(r + 6\sqrt{2}e\sqrt{\frac{K\log(7p^2/\delta)}{n}}) + 12\sqrt{2}e\tau^{-1}\sigma\sqrt{\frac{K\log(7p^2/\delta)}{n}}.$$

Solving the above inequality (notice that $7p^2/\delta < 9p^2/\delta^2$ and $\rho > 1$) completes the proof. $\qquad\square$

The requirement that $\max_{i \ne j} |\Sigma_{ij}| < 1/(\rho s r)$ or the necessary and sufficient condition that $\Sigma$ is RDD strictly constrains the correlation structure of $X$, causing the difficulty for $SIS$ to be strong screening consistent. For $HOLP$ we instead have the following result.

**Lemma 3.** *Let $\Phi = A_{HOLP}X$. Assume $p > c_0 n$ for some $c_0 > 1$, then for any $C > 0$ there exists some $0 < c_1 < 1 < c_2$ and $c_3 > 0$ such that for any $t > 0$ and any $i \in Q, j \ne i$, we have*

$$P\left(|\Phi_{ii}| < c_1\kappa^{-1}\frac{n}{p}\right) \le 2e^{-Cn}, \quad P\left(|\Phi_{ii}| > c_2\kappa\frac{n}{p}\right) \le 2e^{-Cn}$$

*and*

$$P\left(|\Phi_{ij}| > c_4\kappa t\frac{\sqrt{n}}{p}\right) \le 5e^{-Cn} + 2e^{-t^2/2},$$

*where $c_4 = \frac{\sqrt{c_2(c_0 - c_1)}}{\sqrt{c_3(c_0 - 1)}}$.*

*Proof.* The proof of Lemma 3 relies heavily on previous results for the Stiefel Manifold provided in the supplementary materials. We only sketch the basic idea here and leave the complete proof to the supplementary materials. Defining $H = X^T(XX^T)^{-1/2}$, then we have $\Phi = HH^T$ and $H$ follows the Matrix Angular Central Gaussian (MACG) with covariance $\Sigma$. The diagonal terms of $HH^T$ can be bounded similarly via the Johnson-Lindenstrauss lemma, by using the fact that $HH^T = \Sigma^{1/2}U(U^T\Sigma U)^{-1}U\Sigma$, where $U$ is a $p \times n$ random projection matrix. Now for off-diagonal terms, we decompose the Stiefel manifold as $H = (G(H_2)H_1 \ H_2)$, where $H_1$ is a $(p - n + 1) \times 1$ vector, $H_2$ is a $p \times (n - 1)$ matrix and $G(H_2)$ is chosen so that $(G(H_2) \ H_2) \in \mathcal{O}(p)$, and show that $H_1$ follows Angular Central Gaussian (ACG) distribution with covariance $G(H_2)^T\Sigma G(H_2)$ conditional on $H_2$. It can be shown that $e_2 HH^T e_1 \overset{(d)}{=} e_2 G(H_2)H_1 | e_1^T H_2 = 0$. Let $t_1^2 = e_1^T HH^T e_1$, then $e_1^T H_2 = 0$ is equivalent to $e_1^T G(H_2)H_1 = t_1$, and we obtain the desired coupling distribution as $e_2^T HH^T e_1 \overset{(d)}{=} e_2^T G(H_2)H_1 | e_1^T G(H_2)H_1 = t_1$. Using the normal representation of $ACG(\Sigma)$, i.e., if $x = (x_1, \cdots, x_p) \sim N(0, \Sigma)$, then $x/\|x\| \sim ACG(\Sigma)$, we can write $G(H_2)H_1$ in terms of normal variables and then bound all terms using concentration inequalities. $\qquad\square$

Lemma 3 quantifies the entries of the screening matrix for $HOLP$. As illustrated in the lemma, regardless of the covariance $\Sigma$, diagonal terms of $\Phi$ are always $O(\frac{n}{p})$ and the off-diagonal terms are $O(\frac{\sqrt{n}}{p})$. Thus, with $n \ge O(s^2)$, $\Phi$ is likely to satisfy the RDD condition with large probability. For the noise vector we have the following result.

**Lemma 4.** *Let $\eta = A_{HOLP}\epsilon$. Assume $p > c_0 n$ for some $c_0 > 1$, then for any $C > 0$ there exist the same $c_1, c_2, c_3$ as in Lemma 3 such that for any $t > 0$ and $i \in Q$,*

$$P\left(|\eta_i| \ge \frac{2\sigma\sqrt{c_2}\kappa t}{1 - c_0^{-1}}\frac{\sqrt{n}}{p}\right) < 4e^{-Cn} + 2e^{-t^2/2},$$

*if $n \ge 8C/(c_0 - 1)^2$.*

The proof is almost identical to Lemma 2 and is provided in the supplementary materials. The following theorem results after combining Lemma 3 and 4.

**Theorem 6.** *Assume $p > c_0 n$ for some $c_0 > 1$. For any $\delta > 0$, if the sample size satisfies*

$$n > \max\left\{2C'\kappa^4(\rho s + \sigma/\tau)^2 \log(3p/\delta),\; \frac{8C}{(c_0 - 1)^2}\right\}, \tag{6}$$

*where $C' = \max\{\frac{4c_4^2}{c_1^2}, \frac{4c_2}{c_1^2(1-c_0^{-1})^2}\}$ and $c_1, c_2, c_3, c_4, C$ are the same constants defined in Lemma 3, then with probability at least $1 - \delta$, $\Phi = A_{HOLP}X - 2\tau^{-1}\|A_{HOLP}\epsilon\|_\infty I_p$ is restricted diagonally dominant with sparsity $s$ and $C_0 \geq \rho$. This implies HOLP is screening consistent for any $\beta \in \mathcal{B}_\tau(s, \rho)$.*

*Proof.* Notice that if

$$\min_i |\Phi_{ii}| > 2s\rho \max_{ij} |\Phi_{ij}| + 2\tau^{-1}\|X^T(XX^T)^{-1}\epsilon\|_\infty, \tag{7}$$

then the proof is complete because $\Phi - 2\tau^{-1}\|X^T(XX^T)^{-1}\epsilon\|_\infty$ is already a restricted diagonally dominant matrix. Let $t = \sqrt{Cn}/\nu$. The above equation then requires

$$c_1\kappa^{-1}\frac{n}{p} - \frac{2c_4\sqrt{C}\kappa s\rho}{\nu}\frac{n}{p} - \frac{2\sigma\sqrt{c_2 C}\kappa t}{(1-c_0^{-1})\tau\nu}\frac{n}{p} = \left(c_1\kappa^{-1} - \frac{2c_4\sqrt{C}\kappa s\rho}{\nu} - \frac{2\sigma\sqrt{c_2 C}\kappa}{(1-c_0^{-1})\tau\nu}\right)\frac{n}{p} > 0,$$

which implies that

$$\nu > \frac{2c_4\sqrt{C}\kappa^2\rho s}{c_1} + \frac{2\sigma\sqrt{c_2 C}\kappa^2}{c_1(1-c_0^{-1})\tau} = C_1\kappa^2\rho s + C_2\kappa^2\tau^{-1}\sigma > 1,$$

where $C_1 = \frac{2c_4\sqrt{C}}{c_1}$, $C_2 = \frac{2\sqrt{c_2 C}}{c_1(1-c_0^{-1})}$. Therefore, taking union bounds on all matrix entries, we have

$$P\left(\{(7) \text{ does not hold}\}\right) < (p + 5p^2)e^{-Cn} + 2p^2 e^{-Cn/\nu} < (7 + \frac{1}{n})p^2 e^{-Cn/\nu^2},$$

where the second inequality is due to the fact that $p > n$ and $\nu > 1$. Now for any $\delta > 0$, (7) holds with probability at least $1 - \delta$ if

$$n \geq \frac{\nu^2}{C}\left(\log(7 + 1/n) + 2\log p - \log\delta\right),$$

which is satisfied provided (noticing $\sqrt{8} < 3$) $n \geq \frac{2\nu^2}{C}\log\frac{3p}{\delta}$. Now pushing $\nu$ to the limit gives (6), the precise condition we need. $\square$

There are several interesting observations on equation (5) and (6). First, $(\rho s + \sigma/\tau)^2$ appears in both expressions. We note that $\rho s$ evaluates the sparsity and the diversity of the signal $\beta$ while $\sigma/\tau$ is closely related to the signal-to-noise ratio. Furthermore, $HOLP$ relaxes the correlation constraint $r < 1/(2\rho s)$ or the covariance constraint ($\Sigma$ is RDD) with the conditional number constraint. Thus for any $\Sigma$, as long as the sample size is large enough, strong screening consistency is assured. Finally, $HOLP$ provides an example to satisfy the RDD condition in answer to the question raised in Section 4.

# 6 Concluding remarks

This article studies and establishes a necessary and sufficient condition in the form of restricted diagonally dominant screening matrices for strong screening consistency of a linear screener. We verify the condition for both $SIS$ and $HOLP$ under random designs. In addition, we show a close relationship between RDD and the IC, highlighting the difficulty of using SIS in screening for arbitrarily correlated predictors. For future work, it is of interest to see how linear screening can be adapted to compressed sensing [20] and how techniques such as preconditioning [21] can improve the performance of marginal screening and variable selection.

**Acknowledgments** This research was partly support by grant NIH R01-ES017436 from the National Institute of Environmental Health Sciences.

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
