[Supplementary Material]

# On the consistency theory of high dimensional variable screening –supplemantary materials

**Xiangyu Wang**
Dept. of Statistical Science
Duke University, USA
xw56@stat.duke.edu

**Chenlei Leng**
Dept. of Statistics
University of Warwick, UK
C.Leng@warwick.ac.uk

**David B. Dunson**
Dept. of Statistical Science
Duke University, USA
dunson@stat.duke.edu

## 1 Proofs for Section 3

In this section, we prove the two theorems in Section 3.

***Proof of Theorem 1***. If $\Phi$ is restricted diagonally dominant with sparsity $s$ and $C_0 \geq \rho$, we have for any $I \subseteq Q$ and $|I| \leq s - 1$,

$$\Phi_{ii} > \rho \max \left\{ \sum_{j \in I} |\Phi_{ij} + \Phi_{kj}|, \ \sum_{j \in I} |\Phi_{ij} - \Phi_{kj}| \right\} + |\Phi_{ik}| \quad \forall k \neq i \in Q \setminus I.$$

Recall $\hat{\beta} = \Phi \beta$. Suppose $S$ is the index set of non-zero predictors. For any $i \in S, k \notin S$, of we fix $I = S \setminus \{i\}$, we have

$$
\begin{aligned}
|\hat{\beta}_i| &= |\Phi_{ii}\beta_i + \sum_{j \in I} \Phi_{ij}\beta_j| \geq |\beta_i|(\Phi_{ii} + \sum_{j \in I} \frac{\beta_j}{\beta_i} \Phi_{ij}) \\
&= |\beta_i|(\Phi_{ii} + \sum_{j \in I} \frac{\beta_j}{\beta_i}(\Phi_{ij} + \Phi_{kj}) + \Phi_{ki} - \sum_{j \in I} \frac{\beta_j}{\beta_i}\Phi_{kj} - \Phi_{ki}) \\
&> -|\beta_i|(\sum_{j \in I} \frac{\beta_j}{\beta_i}\Phi_{kj} + \Phi_{ki}) = -\frac{|\beta_i|}{\beta_i}(\sum_{j \in I} \beta_j \Phi_{kj} + \beta_i \Phi_{ki}) \\
&= -sign(\beta_i) \cdot \hat{\beta}_k.
\end{aligned}
$$

Similarly we have

$$
\begin{aligned}
|\hat{\beta}_i| &= |\Phi_{ii}\beta_i + \sum_{j \in I} \Phi_{ij}\beta_j| \geq |\beta_i|(\Phi_{ii} + \sum_{j \in I} \frac{\beta_j}{\beta_i} \Phi_{ij}) \\
&= |\beta_i|(\Phi_{ii} + \sum_{j \in I} \frac{\beta_j}{\beta_i}(\Phi_{ij} - \Phi_{kj}) - \Phi_{ki} + \sum_{j \in I} \frac{\beta_j}{\beta_i}\Phi_{kj} + \Phi_{ki}) \\
&> |\beta_i|(\sum_{j \in I} \frac{\beta_j}{\beta_i}\Phi_{kj} + \Phi_{ki}) = sign(\beta_i) \cdot \hat{\beta}_k.
\end{aligned}
$$

Therefore, whatever value $sign(\beta_i)$ is, it always holds that $|\hat{\beta}_i| > |\hat{\beta}_k|$. Since this result is true for any $i \in S, k \notin S$, we have

$$\min_{i \in S} |\hat{\beta}_i| > \max_{k \notin S} |\hat{\beta}_k|.$$

To prove the sign consistency for non-zero coefficients, notice that for $i \in S$,

$$\Phi_{ii} > \rho(\sum_{j \in I} |\Phi_{ij} + \Phi_{kj}| + \sum_{j \in I} |\Phi_{ij} - \Phi_{kj}|)/2 \geq \rho \sum_{j \in I} |\Phi_{ij}|.$$

Thus,

$$\hat{\beta}_i \beta_i = \Phi_{ii}\beta_i^2 + \sum_{j\in I}\Phi_{ij}\beta_j\beta_i = \beta_i^2(\Phi_{ii} + \sum_{j\in I}\frac{\beta_j}{\beta_i}\Phi_{ij}) > 0.$$

On the other hand, if $\hat{\beta}$ is screening consistent, i.e., $|\hat{\beta}_i| \geq |\hat{\beta}_k|$ and $\hat{\beta}_i\beta_i \geq 0$, we can construct $S = I \cup \{i\}$ for any fixed $i, k, I$. Without loss of generality, we assume $\Phi_{ik} \geq 0$. If we select $\beta$ such that $\beta_i > 0$, then the strong screening consistency implies $\hat{\beta}_i > \hat{\beta}_k$ and $\hat{\beta}_i > -\hat{\beta}_k$. From $\hat{\beta}_i > \hat{\beta}_k$ we have

$$\Phi_{ii}\beta_i + \sum_{j\in I}\Phi_{ij}\beta_j > \sum_{j\in I}\Phi_{kj}\beta_j + \Phi_{ki}\beta_i.$$

By rearranging terms and selecting $\beta \in \mathcal{B}(s,\rho)$ as $\beta_i = 1, \beta_j = -\rho \cdot sign(\Phi_{ij} - \Phi_{kj}), j \in S$ we have

$$\Phi_{ii} > -\sum_{j\in I}(\Phi_{ij} - \Phi_{kj})\beta_j + \Phi_{ki} = \rho\sum_{j\in I}|\Phi_{ij} - \Phi_{kj}| + |\Phi_{ki}|.$$

Following the same argument on $\hat{\beta}_i \geq -\hat{\beta}_k$ with a choice of $\beta_i = 1, \beta_j = -\rho \cdot sign(\Phi_{ij} + \Phi_{kj}), j \in S$ we have

$$\Phi_{ii} > \rho\sum_{j\in I}|\Phi_{ij} + \Phi_{kj}| + |\Phi_{ki}|.$$

This concludes the proof. $\qquad\square$

***Proof of Theorem 2.*** Proof of Lemma 3 follows almost the same as the sufficiency part of Theorem 1. Notice that now the definition of $\hat{\beta}$ becomes

$$\hat{\beta} = X^T(XX^T)^{-1}X\beta + X^T(XX^T)^{-1}\epsilon.$$

If the condition holds, i.e., for any $i \in S$, $I = S \setminus \{i\}$ and $k \notin S$, we have

$$\Phi_{ii} > \rho\max\left\{\sum_{j\in I}|\Phi_{ij} + \Phi_{kj}|, \sum_{j\in I}|\Phi_{ij} - \Phi_{kj}|\right\} + |\Phi_{ik}| + 2\tau^{-1}\|X^T(XX^T)^{-1}\epsilon\|_\infty.$$

Defining $\eta = X^T(XX^T)^{-1}\epsilon$, we have for any $i \in S$,

$$|\hat{\beta}_i| = |\Phi_{ii}\beta_i + \sum_{j\in I}\Phi_{ij}\beta_j + \eta_i| \geq |\beta_i|(\Phi_{ii} + \sum_{j\in I}\frac{\beta_j}{\beta_i}\Phi_{ij} + \beta_i^{-1}\eta_i)$$

$$= |\beta_i|(\Phi_{ii} + \sum_{j\in I}\frac{\beta_j}{\beta_i}(\Phi_{ij} + \Phi_{kj}) + \Phi_{ki} + \beta_i^{-1}(\eta_i + \eta_k) - \sum_{j\in I}\frac{\beta_j}{\beta_i}\Phi_{kj} - \Phi_{ki} - \beta_i^{-1}\eta_k)$$

$$> -|\beta_i|(\sum_{j\in I}\frac{\beta_j}{\beta_i}\Phi_{kj} + \Phi_{ki} + \beta_i^{-1}\eta_k) = -\frac{|\beta_i|}{\beta_i}(\sum_{j\in I}\beta_j\Phi_{kj} + \beta_i\Phi_{ki} + \eta_k)$$

$$= -sign(\beta_i) \cdot \hat{\beta}_k,$$

Similarly, we can prove $|\hat{\beta}_i| > sign(\beta_i) \cdot \hat{\beta}_k$, and thus $|\hat{\beta}_i| > |\hat{\beta}_k|$, which implies that

$$\min_{i\in S}|\hat{\beta}_i| > \max_{k\notin S}|\hat{\beta}_k|.$$

The weak sign consistency is established since

$$\hat{\beta}_i\beta_i = \Phi_{ii}\beta_i^2 + \sum_{j\in I}\Phi_{ij}\beta_j\beta_i + \eta_i\beta_i = \beta_i^2(\Phi_{ii} + \sum_{j\in I}\frac{\beta_j}{\beta_i}\Phi_{ij} + \beta_i^{-1}\eta_i) > 0,$$

for any $\beta_i \neq 0$.

The tightness of this theorem is given by the case when $\epsilon = 0$, for which the condition has already been shown to be necessary and sufficient in Theorem 1. $\qquad\square$

## 2 Proofs for Section 4

In this section, we prove results from Section 4 that are not covered in the main article.

***Proof of Corollary 1***. Letting $I \subseteq Q, |I| \leq s - 1$, we have for any $i \neq k \in Q \setminus I$,

$$\Phi_{ii} - \frac{1}{c} \max \left\{ \sum_{j \in I} |\Phi_{ij} + \Phi_{kj}|, \sum_{j \in I} |\Phi_{ij} - \Phi_{kj}| \right\} + |\Phi_{ik}| \geq 1 - \frac{1}{c} \left( 2(s-1)\frac{c}{2s} + \frac{c}{2s} \right) = \frac{1}{2s} > 0.$$

This completes the proof for the first case.

Now for the second case, notice that the sum of an entire row (except the diagonal term) can be bounded by $\sum_{j \neq i} |\Phi_{ij}| < 2 \sum_{k=1}^{\infty} r^k < \frac{2r}{1-r}$. Therefore, we have

$$\Phi_{ii} - \frac{(1-r)^2}{4r} \max \left\{ \sum_{j \in I} |\Phi_{ij} + \Phi_{kj}|, \sum_{j \in I} |\Phi_{ij} - \Phi_{kj}| \right\} - |\Phi_{ik}| > 1 - \frac{(1-r)^2}{2r} \sum_{j \neq i} |\Phi_{ij}| - r = 0.$$

$\square$

***Proof of Theorem 3***. First, from RDD to IC: Without loss of generality, we assume $S = \{1, 2\}$. For any $k \in Q \setminus S$, we have

$$\left| [\Phi_{k1} \ \Phi_{k2}] \Phi_{S, S}^{-1} sign(\beta_S) \right| = \left| \frac{sign(\beta_1)(\Phi_{k1} - \Phi_{12}\Phi_{k2}) + sign(\beta_2)(-\Phi_{12}\Phi_{k1} + \Phi_{k2})}{1 - \Phi_{12}^2} \right|.$$

The r.h.s. becomes $|\Phi_{k1} + \Phi_{k2}|(1 - \Phi_{12})/(1 - \Phi_{12}^2)$ when $sign(\beta_1) = sign(\beta_2)$ and $|\Phi_{k1} - \Phi_{k2}|(1 + \Phi_{12})/(1 - \Phi_{12}^2)$ when $sign(\beta_1) = -sign(\beta_2)$. In either case we have

$$\left| [\Phi_{k1} \ \Phi_{k2}] \Phi_{S, S}^{-1} sign(\beta_S) \right| \leq \frac{\max \left\{ |\Phi_{1k} + \Phi_{2k}|, |\Phi_{1k} - \Phi_{2k}| \right\}}{1 - r} < \frac{\rho^{-1}}{1 - r}.$$

Second, from IC to RDD: Let $I \subseteq Q, |I| = 1$ and $i \neq k \in Q \setminus I$. Without loss of generality, we assume $i = 1, k = 2$, and we construct $S = \{1, 2\}$. Now for any $j \in I$, using the same formula as shown above, we have

$$1 - \theta \geq \left| [\Phi_{j1} \ \Phi_{j2}] \Phi_{S, S}^{-1} sign(\beta_S) \right| = \left| \frac{sign(\beta_1)(\Phi_{j1} - \Phi_{12}\Phi_{j2}) + sign(\beta_2)(-\Phi_{12}\Phi_{j1} + \Phi_{j2})}{1 - \Phi_{12}^2} \right|.$$

Using the same result on the r.h.s., i.e., it becomes $|\Phi_{k1} + \Phi_{k2}|(1 - \Phi_{12})/(1 - \Phi_{12}^2)$ when $sign(\beta_1) = sign(\beta_2)$ and $|\Phi_{k1} - \Phi_{k2}|(1 + \Phi_{12})/(1 - \Phi_{12}^2)$ when $sign(\beta_1) = -sign(\beta_2)$, we have for any $j \in I$ that

$$\max \left\{ |\Phi_{1j} + \Phi_{2j}|, |\Phi_{1j} - \Phi_{2j}| \right\} \leq (1 - \theta)(1 + r).$$

As a result, we have

$$\sum_{j \in I} \max \left\{ |\Phi_{1j} + \Phi_{2j}|, |\Phi_{1j} - \Phi_{2j}| \right\} < (1 - \theta)(1 + r) < (1 - \theta)\frac{1 + r}{1 - r} \left( \Phi_{11} - |\Phi_{12}| \right),$$

which implies

$$\Phi_{11} > \frac{1}{1 - \theta} \frac{1 - r}{1 + r} \sum_{j \in I} \max \left\{ |\Phi_{1j} + \Phi_{2j}|, |\Phi_{1j} - \Phi_{2j}| \right\} + |\Phi_{12}|.$$

$\square$

***Proof of Theorem 4***. We just need to check (4). We prove the absolute value of the first coordinate of $C_{S^c, S} C_{S, S}^{-1} \cdot sign(\beta_S)$ is less than one, and the rest just follow the same argument. From the

condition we know $C = X^T X/n$ is restricted diagonally dominant. Then equation (3) implies that for any $I \subseteq Q$ with $|I| = s$, we have for any $k \notin I$,

$$\rho \sum_{i \in I} |C_{ki}| < 1.$$

Now for any $S \subseteq Q$ with $|S| = s$, we choose $I = S$ and let $\alpha^T$ be the first row of $C_{S^c, S} = X_{S^c}^T X_S/n$, we have

$$|\alpha^T (X_S^T X_S/n)^{-1} sign(\beta_S)| \leq \|\alpha\|_2 \|sign(\beta_S)\|_2 \mu^{-1}.$$

Because $\rho \sum_{i=1}^{s} |\alpha_i| < 1$, we have

$$\rho^2 \sum_{i=1}^{s} \alpha_i^2 < \rho^2 (\sum_{i=1}^{s} |\alpha_i|)^2 < 1,$$

which implies that

$$|\alpha^T (X_S^T X_S/n)^{-1} sign(\beta_S)| \leq \rho^{-1} \sqrt{s} \mu^{-1} = \frac{\sqrt{s}}{\rho\mu} < 1.$$

$\square$

## 3 Proofs for Section 6 (SIS)

Proofs in Section 6 are divided into two parts. In this section, we provide the proofs related to SIS, and leave those pertaining to HOLP to the next section. The proof requires the following proposition,

**Proposition 1.** *Assume $X_i \sim \mathcal{X}^2(1), i = 1, 2, \cdots, n$, where $\mathcal{X}^2(1)$ is the chi-square distribution with one degree of freedom. Then for any $t > 0$, we have*

$$P(|\frac{\sum_{i=1}^{n} X_i}{n} - 1| \geq t) \leq 2 \exp\left\{ - \min\left( \frac{t^2 n}{8e^2 K}, \frac{tn}{2eK} \right) \right\},$$

*where $K = \|\mathcal{X}^2(1) - 1\|_{\psi_1}$. Alternatively, for any $C > 0$, there exists some $0 < c_3 < 1 < c_4$ such that,*

$$P(\frac{\sum_{i=1}^{n} X_i}{n} \leq c_3) \leq e^{-Cn}, \tag{1}$$

*and*

$$P(\frac{\sum_{i=1}^{n} X_i}{n} \geq c_4) \leq e^{-Cn}.$$

*Proof.* It is a direct application of Proposition 5.16 in [1]. Notice that in the proof of Proposition 5.16 we have $C = 2e^2$ and $c = e/2$ for $\mathcal{X}^2(1) - 1$. $\square$

***Proof of Lemma 1.*** For diagonal term we have for any $i \in \{1, 2, \cdots, p\}$

$$\Phi_{ii} - \Sigma_{ii} = \frac{\sum_{k=1}^{n} x_{ik}^2}{n} - 1,$$

where $x_{ik}, k = 1, 2, \cdots, n$'s are $n$ iid standard normal random variables. Using Proposition 1, we have for any $t > 0$,

$$P\left( |\Phi_{ii} - \Sigma_{ii}| \geq t \right) \leq 2 \exp\left\{ - \min\left( \frac{t^2 n}{8e^2 K}, \frac{tn}{2eK} \right) \right\}. \tag{2}$$

For the off-diagonal term, we have for any $i \neq j$,

$$\Phi_{ij} - \Sigma_{ij} = \frac{\sum_{k=1}^{n} x_{ik} x_{jk}}{n} - \Sigma_{ij}$$

$$= \frac{\sum_{k=1}^{n} (x_{ik} + x_{jk})^2}{2n} - \frac{\sum_{k=1}^{n} x_{ik}^2}{2n} - \frac{\sum_{k=1}^{n} x_{jk}^2}{2n} - \Sigma_{ij}$$

$$= \frac{1}{2}\left( \frac{\sum_{k=1}^{n} (x_{ik} + x_{jk})^2}{n} - (2 + 2\Sigma_{ij}) \right) - \frac{1}{2}\left( \frac{\sum_{k=1}^{n} x_{ik}^2}{n} - 1 \right) - \frac{1}{2}\left( \frac{\sum_{k=1}^{n} x_{jk}^2}{n} - 1 \right).$$

Notice that $x_{ik} + x_{jk} \sim N(0, 2 + 2\Sigma_{ij})$. Hence the three terms in the above equation can be bounded using the same inequality before, i.e., for any $t > 0$,

$$P\left(|\Phi_{ij} - \Sigma_{ij}| \geq (2 + \Sigma_{ij})t\right) \leq 6\exp\left\{-\min\left(\frac{t^2 n}{8e^2}, \frac{tn}{2e}\right)\right\}.$$

Clearly, we have $\Sigma_{ij} \leq \sqrt{\Sigma_{ii}}\sqrt{\Sigma_{jj}} \leq 1$. Therefore, we have

$$P\left(|\Phi_{ij} - \Sigma_{ij}| \geq t\right) \leq 6\exp\left\{-\min\left(\frac{t^2 n}{72e^2 K}, \frac{tn}{6eK}\right)\right\}.$$

$\square$

***Proof of Lemma 2.*** The proof is essentially the same for proving the off diagonal terms of $\Phi$ as in Lemma 1. The only difference is that $E(\Phi_{ij}) = \Sigma_{ij}$ while $E(X\epsilon) = 0$. Note

$$\eta_i/\sigma = \frac{\sum_{k=1}^n x_{ik}\epsilon_k/\sigma}{n} = \frac{\sum_{k=1}^n (x_{ik} + \epsilon_k/\sigma)^2}{2n} - \frac{\sum_{k=1}^n x_{ik}^2}{2n} - \frac{\sum_{k=1}^n \epsilon_k^2/\sigma^2}{2n}.$$

Using Proposition 1, we have

$$P\left(|\eta_i/\sigma| \geq t\right) \leq 6\exp\left\{-\min\left(\frac{t^2 n}{72e^2 K}, \frac{tn}{6eK}\right)\right\}.$$

$\square$

Now we turn to the proof of Theorem 5.

***Proof of Theorem 5.*** Taking union bound on the results from Lemma 1 and 2, we have for any $t > 0$,

$$P\left(\min_{i \in Q} \Phi_{ii} \leq 1 - t\right) \leq 2p\exp\left\{-\min\left(\frac{t^2 n}{8e^2 K}, \frac{tn}{2eK}\right)\right\},$$

$$P\left(\max_{i \neq j} |\Phi_{ij}| \geq r + t\right) \leq 6(p^2 - p)\exp\left\{-\min\left(\frac{t^2 n}{72e^2 K}, \frac{tn}{6eK}\right)\right\},$$

and

$$P\left(\max_{i \in Q} |\eta_i| \geq \sigma t\right) \leq 6p\exp\left\{-\min\left(\frac{t^2 n}{72e^2 K}, \frac{tn}{6eK}\right)\right\}.$$

Thus, when $p > 2$ we have

$$P\left(\min_{i \in Q} \Phi_{ii} \leq 1 - t \text{ or } \max_{i \neq j} |\Phi_{ij}| \geq r + t \text{ or } \max_{i \in Q} |\eta_i| \geq \sigma t\right) \leq 7p^2\exp\left\{-\min\left(\frac{t^2 n}{72e^2 K}, \frac{tn}{6eK}\right)\right\}.$$

In other words, for any $\delta > 0$, when $n \geq K\log(7p^2/\delta)$, with probability at least $1 - \delta$, we have

$$\min_{i \in Q} \Phi_{ii} \geq 1 - 6\sqrt{2}e\sqrt{\frac{K\log(7p^2/\delta)}{n}}, \quad \max_{i \neq j} |\Phi_{ij}| \leq r + 6\sqrt{2}e\sqrt{\frac{K\log(7p^2/\delta)}{n}},$$

and

$$\max_{i \in Q} |\eta_i| \leq 6\sqrt{2}e\sigma\sqrt{\frac{K\log(7p^2/\delta)}{n}}.$$

A sufficient condition for $\Phi$ to be restricted diagonally dominant is that

$$\min_i \Phi_{ii} > 2\rho s \max_{i \neq j} |\Phi_{ij}| + 2\tau^{-1} \max_i |\eta_i|.$$

Plugging in the values and solving the inequality, we have (notice that $7p^2/\delta < 9p^2/\delta^2$) $\Phi$ is RDD as long as

$$n > 144K\left(\frac{1 + 2\rho s + 2\sigma/\tau}{1 - 2\rho sr}\right)^2 \log(3p/\delta).$$

This completes the proof.

$\square$

# 4 Proofs for Section 6 (HOLP)

In this section we prove Lemma 3, 4 and Theorem 5. Several propositions and lemmas are needed for establishing the whole theory. We list all prerequisite results without proofs but provide readers references for complete proofs.

Let $P \in \mathcal{O}(p)$ be a $p \times p$ orthogonal matrix from the orthogonal group $\mathcal{O}(p)$. Let $H$ denote the first $n$ columns of $P$. Then $H$ is in the Stiefel manifold [2]. In general, the Stiefel manifold $V_{n,p}$ is the space whose points are $n$-frames in $\mathcal{R}^p$ represented as the set of $p \times n$ matrices $X$ such that $X^T X = I_n$. Mathematically, we can write

$$V_{n,p} = \{X \in R^{p \times n} : X^T X = I_n\}.$$

There is a natural measure $(dX)$ called Haar measure on the Stiefel manifold, invariant under both right orthogonal and left orthogonal transformations. We standardize it to obtain a probability measure as $[dX] = (dX)/V(n,p)$, where $V(n,p) = 2^n \pi^{np/2}/\Gamma_n(1/2p)$.

**Lemma 1.** *[2, Page 41-44] Supposed that a $p \times n$ random matrix $Z$ has the density function of the form*

$$f_Z(Z) = |\Sigma|^{-n/2} g(Z^T \Sigma^{-1} Z),$$

*which is invariant under the right-orthogonal transformation of $Z$, where $\Sigma$ is a $p \times p$ positive definite matrix. Then its orientation $H_z = Z(Z^T Z)^{-1/2}$ has the matrix angular central Gaussian distribution (MACG) with a probability density function*

$$MACG(\Sigma) = |\Sigma|^{-n/2} |H_z^T \Sigma^{-1} H_z|^{-p/2}.$$

*In particular, if $Z$ is a $p \times n$ matrix whose distribution is invariant under both the left- and right-orthogonal transformations, then $H_Y$, with $Y = BZ$ for $BB^T = \Sigma$, has the $MACG(\Sigma)$ distribution.*

When $n = 1$, the MACG distribution becomes the angular central Gaussian distribution, a description of the multivariate Gaussian distribution on the unit sphere [3].

**Lemma 2.** *[2, Page 70, Decomposition of the Stiefel manifold] Let $H$ be a $p \times n$ random matrix on $V_{n,p}$, and write*

$$H = (H_1 \ H_2),$$

*with $H_1$ being a $p \times q$ matrix where $0 < q < n$. Then we can write*

$$H_2 = G(H_1)U_1,$$

*where $G(H_1)$ is any matrix chosen so that $(H_1 \ G(H_1)) \in \mathcal{O}(p)$; as $H_2$ runs over $V_{n-q,p}$, $U_1$ runs over $V_{n-q,p-q}$ and the relationship is one to one. The differential form $[dH]$ for the normalized invariant measure on $V_{n,p}$ is decomposed as the product*

$$[dH] = [dH_1][dU_1]$$

*of those $[dH_1]$ and $[dU_1]$ on $V_{q,p}$ and $V_{n-q,p-q}$, respectively.*

**Lemma 3.** *[Lemma 4 in [4]]Let $U$ be uniformly distributed on the Stiefel manifold $V_{n,p}$. Then for any $C > 0$, there exist $c_1', c_2'$ with $0 < c_1' < 1 < c_2'$, such that*

$$P\left(e_1^T U U^T e_1 < c_1' \frac{n}{p}\right) \leq 2e^{-Cn},$$

*and*

$$P\left(e_1^T U U^T e_1 > c_2' \frac{n}{p}\right) \leq 4e^{-Cn}.$$

Some of our proof requires concentration properties of a random Gaussian matrix and $\mathcal{X}_1^2$ random variables. For a Wigner matrix, we have the following result.

**Lemma 4.** *Assume $Z$ is a $n \times p$ matrix with $p > c_0 n$ for some $c_0 > 1$. Each entry of $Z$ follows a Gaussian distribution with mean zero and variance one and are independent. Then for any $t > 0$, with probability at least $1 - 2\exp(-t^2/2)$, we have*

$$(1 - c_0^{-1} - t/p)^2 \leq \lambda_{min}(ZZ^T/p) < \lambda_{max}(ZZ^T/p) \leq (1 + c_0^{-1} + t/p)^2.$$

*For any $C > 0$, taking $t = \sqrt{2Cn}$, we have with probability $1 - 2\exp(-Cn/2)$,*

$$(1 - c_0^{-1} - \frac{\sqrt{2C}}{c_0\sqrt{n}})^2 \leq \lambda_{min}(ZZ^T/p) \leq (1 + c_0^{-1} + \frac{\sqrt{2C}}{c_0\sqrt{n}})^2.$$

*Proof.* This is essentially Corollary 5.35 in [1]. □

The conditional number of $\Sigma$ is controled by $\kappa$, which simulaneously controls the largest and the smallest eigenvalues.

**Proposition 2.** *Assume the conditional number of $\Sigma$ is $\kappa$ and $\Sigma_{ii} = 1$ for $i = 1, 2, \cdots, p$, then we have*

$$\lambda_{min}(\Sigma) \geq \kappa^{-1} \qquad and \qquad \lambda_{max}(\Sigma) \leq \kappa.$$

*Proof.* Notice that $p = tr(\Sigma) = \sum_{i=1}^{p} \lambda_i$. Therefore, we have

$$p/\lambda_{max} \geq p\kappa^{-1} \quad and \quad p/\lambda_{min}(\Sigma) \leq p\kappa,$$

which completes the proof. □

Now we prove the main results for HOLP.

***Proof of Lemma 3.*** Consider a transformed $n \times p$ random matrix $Z = X\Sigma^{-1/2}$, which, by definition, follows standard multivariate Gaussian. Consider its SVD decomposition,

$$Z = VDU^T,$$

where $V \in \mathcal{O}(n)$, $D$ is a diagonal matrix and $U$ is a $p \times n$ random matrix belonging to the Stiefel manifold $V_{n,p}$. With such notion, we can rewrite the projection matrix as

$$X^T(XX^T)^{-1}X = \Sigma^{1/2}U(U^T\Sigma U)^{-1}U^T\Sigma^{1/2} = HH^T,$$

where $H = \Sigma^{1/2}U(U^T\Sigma U)^{-1/2}$ and $H \in V_{n,p-1}$. Therefore, the two quantities that we are interested in are $\Phi_{ii} = e_i^T HH^T e_i$ (diagonal term) and $\Phi_{ij} = e_i^T HH^T e_j$ (off-diagonal term), where $e_i^T$ is the $p-$dimensional unit vector with the $i^{th}$ coordinate being one. The proof is divided into two parts, where in the first part we consider diagonal terms and the second part takes care of off-diagonal terms.

**Part I:** First, we consider the diagonal term $e_i^T HH^T e_i$. Recall the definition of $H$ and

$$e_i^T HH^T e_i = e_i^T \Sigma^{\frac{1}{2}} U(U^T\Sigma U)^{-1}U^T\Sigma^{\frac{1}{2}}e_i.$$

There always exists some orthogonal matrix $Q$ that rotates the vector $\Sigma^{\frac{1}{2}}e_i$ to the direction of $e_1$, i.e,

$$\Sigma^{\frac{1}{2}}v = \|\Sigma^{\frac{1}{2}}v\|Qe_1.$$

Then we have

$$e_i^T HH^T e_i = \|\Sigma^{\frac{1}{2}}e_i\|^2 e_1^T Q^T U(U^T\Sigma U)^{-1}U^T Qe_1 = \|\Sigma^{\frac{1}{2}}v\|^2 e_1^T \tilde{U}(U^T\Sigma U)^{-1}\tilde{U}e_1,$$

where $\tilde{U} = Q^T U$ is uniformly distributed on $V_{n,p}$, because $U$ is uniformly distributed on $V_{n,p}$ (see discussion in the beginning). Now the magnitude of $e_i^T HH^T e_I$ can be evaluated in two parts. For the norm of the vector $\Sigma^{\frac{1}{2}}v$, we have

$$\lambda_{min}(\Sigma) \leq e_i^T \Sigma e_i = \|\Sigma^{\frac{1}{2}}e)i\|^2 \leq \lambda_{max}(\Sigma), \tag{3}$$

and for the remaining part,

$$e_1^T \tilde{U}(U^T \Sigma U)^{-1} \tilde{U} e_1 \leq \lambda_{max}((U^T \Sigma U)^{-1}) \|\tilde{U} e_1\|^2 \leq \lambda_{min}(\Sigma)^{-1} \|\tilde{U} e_1\|^2,$$

and

$$e_1^T \tilde{U}(U^T \Sigma U)^{-1} \tilde{U} e_1 \geq \lambda_{min}((U^T \Sigma U)^{-1}) \|\tilde{U} e_1\|^2 \geq \lambda_{max}(\Sigma)^{-1} \|\tilde{U} e_1\|^2.$$

Consequently, we have

$$e_i^T H H^T e_i \leq \frac{\lambda_{max}(\Sigma)}{\lambda_{min}(\Sigma)} e_1^T U U^T e_1, \qquad e_i^T H H^T e_i \geq \frac{\lambda_{min}(\Sigma)}{\lambda_{max}(\Sigma)} e_1^T U U^T e_1. \qquad (4)$$

Therefore, following Proposition 3, for any $C > 0$ we have

$$P\left( e_i^T H H^T e_i < c_1' c_4 \kappa^{-1} \frac{n}{p} \right) \leq 2e^{-Cn},$$

and

$$P\left( e_i^T H H^T e_i > c_2' c_4^{-1} \kappa \frac{n^1}{p} \right) \leq 2e^{-Cn}.$$

Denoting $c_1' c_4$ by $c_1$ and $c_2' c_4^{-1}$ by $c_2$, we obtain the equation in Lemma 3.

**Part II:** Second, for off-diagonal terms, although the proof is almost identical to the proof of Lemma 5 in [5], we still provide a complete version here due to the importance of this result.

The proof depends on the decomposition of Stiefel manifold. Without loss of generality, we prove the bound only for $e_2^T H H^T e_1$, then the other off-diagonal terms should follow exactly the same argument. According to Lemma 2, we can decompose $H = (T_1, H_2)$ with $T_1 = G(H_2)H_1$, where $H_2$ is a $p \times (n-1)$ matrix, $H_1$ is a $(p - n + 1) \times 1$ vector and $G(H_2)$ is a matrix such that $(G(H_2), H_2) \in \mathcal{O}(p)$. The invariant measure on the Stiefel manifold can be decomposed as

$$[H] = [H_1][H_2]$$

where $[H_1]$ and $[H_2]$ are Haar measures on $V_{1,n-p+1}, V_{n-1,p}$ (Notice that $q = n - 1$ in this decomposition) respectively. As pointed out before, $H$ has the $MACG(\Sigma)$ distribution, which possesses a density as

$$p(H) \propto |H^T \Sigma^{-1} H|^{-p/2}[dH].$$

Using the identity for matrix determinant

$$\begin{vmatrix} A & B \\ C & D \end{vmatrix} = |A||D - CA^{-1}B| = |D||A - BD^{-1}C|,$$

we have

$$P(H_1, H_2) \propto |H_2^T \Sigma^{-1} H_2|^{-p/2}(T_1^T \Sigma^{-1} T_1 - T_1^T \Sigma^{-1} H_2(H_2^T \Sigma^{-1} H_2)^{-1} H_2^T \Sigma^{-1} T_1)^{-p/2}$$
$$= |H_2^T \Sigma^{-1} H_2|^{-p/2}(H_1^T G(H_2)^T (\Sigma^{-1} - \Sigma^{-1} H_2(H_2^T \Sigma^{-1} H_2)^{-1} H_2^T \Sigma^{-1}) G(H_2) H_1)^{-p/2}$$
$$= |H_2^T \Sigma^{-1} H_2|^{-p/2}(H_1^T G(H_2)^T \Sigma^{-1/2}(I - T_2)\Sigma^{-1/2} G(H_2) H_1)^{-p/2},$$

where $T_2 = \Sigma^{-1/2} H_2(H_2^T \Sigma^{-1} H_2)^{-1} H_2^T \Sigma^{-1/2}$ is an orthogonal projection onto the linear space spanned by the columns of $\Sigma^{-1/2} H_2$. It is easy to verify the following result by using the definition of $G(H_2)$,

$$[\Sigma^{1/2} G(H_2)(G(H_2)^T \Sigma G(H_2))^{-1/2}, \ \Sigma^{-1/2} H_2(H_2^T \Sigma^{-1} H_2)^{-1/2}] \in \mathcal{O}(p),$$

and therefore we have

$$I - T_2 = \Sigma^{1/2} G(H_2)(G(H_2)^T \Sigma G(H_2))^{-1} G(H_2)^T \Sigma^{1/2},$$

which simplifies the density function as

$$P(H_1, H_2) \propto |H_2^T \Sigma^{-1} H_2|^{-p/2}(H_1^T (G(H_2)^T \Sigma G(H_2))^{-1} H_1)^{-p/2}.$$

Now it becomes clear that $H_1|H_2$ follows the Angular Central Gaussian distribution $ACG(\Sigma')$, where

$$\Sigma' = G(H_2)^T \Sigma G(H_2).$$

Next, we relate the target quantity $e_1^T H H^T e_2$ to the distribution of $H_1$. Notice that for any orthogonal matrix $Q \in \mathcal{O}(n)$, we have

$$e_1^T H H^T e_2 = e_1^T H Q Q^T H^T e_2 = e_1^T H' H'^T e_2.$$

Write $H' = HQ = (T_1', H_2')$, where $T_1' = [T_1^{'(1)}, T_1^{'(2)}, \cdots, T_1^{'(p)}]$, $H_2' = [H_2^{'(i,j)}]$. If we choose $Q$ such that the first row of $H_2'$ are all zero (this is possible as we can choose the first column of $Q$ being the first row of $H$ upon normalizing), i.e.,

$$e_1^T H' = [T_1^{'(1)}, \, 0, \cdots, 0] \qquad e_2^T H' = [T_1^{'(2)}, \, H_2^{'(2,1)}, \cdots, \, H_2^{'(2,n-1)}],$$

then immediately we have $e_1^T H H^T e_2 = e_1^T H' H'^T e_2 = T_1^{'(1)} T_1^{'(2)}$. This indicates that

$$e_1^T H H^T e_2 \overset{(d)}{=} \quad T_1^{(1)} T_1^{(2)} \, \Big| \, e_1^T H_2 = 0.$$

As shown at the beginning, $H_1$ follows $ACG(\Sigma')$ conditional on $H_2$. Let $H_1 = (h_1, h_2, \cdots, h_p)^T$ and let $x^T = (x_1, x_2, \cdots, x_{p-n+1}) \sim N(0, \Sigma')$, then we have

$$h_i \overset{(d)}{=} \frac{x_i}{\sqrt{x_1^2 + \cdots + x_{p-n+1}^2}}.$$

Notice that $T_1 = G(H_2) H_1$, a linear transformation on $H_1$. Defining $y = G(H_2) x$, we have

$$T_1^{(i)} \overset{(d)}{=} \frac{y_i}{\sqrt{y_1^2 + \cdots + y_p^2}}, \tag{5}$$

where $y \sim N(0, G(H) \Sigma' G(H)^T)$ is a degenerate Gaussian distribution. This degenerate distribution contains an interesting form. Letting $z \sim N(0, \Sigma)$, we know $y$ can be expressed as $y = G(H) G(H)^T z$. Write $G(H_2)^T$ as $[g_1, g_2]$ where $g_1$ is a $(p - n + 1) \times 1$ vector and $g_2$ is a $(p - n + 1) \times (p - 1)$ matrix, then we have

$$G(H_2) G(H_2)^T = \begin{pmatrix} g_1^T g_1 & g_1^T g_2 \\ g_2^T g_1 & g_2^T g_2 \end{pmatrix}.$$

We can also write $H_2^T = [0_{n-1,1}, h_2]$ where $h_2$ is a $(n - 1) \times (p - 1)$ matrix, and using the orthogonality, i.e., $[H_2 \, G(H_2)][H_2 \, G(H_2)]^T = I_p$, we have

$$g_1^T g_1 = 1, \; g_1^T g_2 = 0_{1,p-1} \quad \text{and} \quad g_2^T g_2 = I_{p-1} - h_2 h_2^T.$$

Because $h_2$ is a set of orthogonal basis in the $p - 1$ dimensional space, $g_2^T g_2$ is therefore an orthogonal projection onto the space $\{h_2\}^\perp$ and $g_2^T g_2 = A A^T$ where $A = g_2^T (g_2 g_2^T)^{-1/2}$ is a $(p - 1) \times (p - n)$ orientation matrix on $\{h_2\}^\perp$. Together, we have

$$y = \begin{pmatrix} 1 & 0 \\ 0 & A A^T \end{pmatrix} z.$$

This relationship allows us to marginalize $y_1$ out with $y$ following a degenerate Gaussian distribution.

We now turn to transform the condition $e_1^T H_2 = 0$ onto constraints on the distribution of $T_1^{(i)}$. Letting $t_1^2 = e_1^T H H^T e_1$, then $e_1^T H_2 = 0$ is equivalent to $T_1^{(1)2} = e_1^T H H^T e_1 = t_1^2$, which implies that

$$e_1^T H H^T e_2 \overset{(d)}{=} \quad T_1^{(1)} T_1^{(2)} \, \Big| \, T_1^{(1)2} = e_1^T H H^T e_1.$$

Because the magnitude of $e_1^T H H^T e_1$ has been obtained in Part I, we can now condition on the value of $e_1^T H H^T e_1$ to obtain the bound on $T_1^{(2)}$. From $T_1^{(1)2} = t_1^2$, we obtain that,

$$(1 - t_1^2) y_1^2 = t_1^2 (y_2^2 + y_3^2 + \cdots + y_p^2). \tag{6}$$

Notice this constraint is imposed on the norm of $\tilde{y} = (y_2, \; y_3, \cdots, y_p)$ and is thus independent of $(y_2/\|\tilde{y}\|, \cdots, y_p/\|\tilde{y}\|)$. Equation (6) also implies that

$$(1 - t_1^2)(y_1^2 + y_2^2 + \cdots + y_p^2) = y_2^2 + y_3^2 + \cdots + y_p^2. \tag{7}$$

Therefore, combining (5) with (6), (7) and integrating $y_1$ out, we have

$$T_1^{(i)} \mid T_1^{(1)} = t_1 \; \overset{(d)}{=} \; \frac{\sqrt{1 - t_1^2} y_i}{\sqrt{y_2^2 + \cdots + y_p^2}}, \qquad i = 2, 3, \cdots, p,$$

where $(y_2, y_3, \cdots, y_p) \sim N(0, AA^T \Sigma_{22} AA^T)$ with $\Sigma_{22}$ being the covariance matrix of $z_2, \cdots, z_p$. To bound the numerator, we use the classical tail bound on the normal distribution as for any $t > 0$, $(\sigma_i = \sqrt{var(y_i)} \leq \sqrt{\lambda_{max}(AA^T \Sigma_{22} AA^T)} \leq \lambda_{max}(\Sigma)^{1/2})$,

$$P(|y_i| > t\sigma_i) = P(|y_i| > t\lambda_{max}^{\frac{1}{2}}(\Sigma)) \leq 2e^{-t^2/2}. \tag{8}$$

For the denominator, letting $\tilde{z} \sim N(0, I_{p-1})$, we have

$$\tilde{y} = AA^T \Sigma_{22}^{1/2} \tilde{z} \quad \text{and} \quad \tilde{y}^T \tilde{y} = \tilde{z}^T \Sigma_{22}^{1/2} AA^T \Sigma_{22}^{1/2} \tilde{z} \overset{(d)}{=} \sum_{i=1}^{p-n} \lambda_i \mathcal{X}_i^2(1),$$

where $\mathcal{X}_i^2(1)$ are iid chi-square random variables and $\lambda_i$ are non-zero eigenvalues of matrix $\Sigma_{22}^{1/2} AA^T \Sigma_{22}^{1/2}$. Here $\lambda_i$'s are naturally upper bounded by $\lambda_{max}(\Sigma)$. To give a lower bound, notice that $\Sigma_{22}^{1/2} AA^T \Sigma_{22}^{1/2}$ and $A\Sigma_{22} A^T$ possess the same set of non-zero eigenvalues, thus

$$\min_i \lambda_i \geq \lambda_{min}(A\Sigma_{22} A^T) \geq \lambda_{min}(\Sigma).$$

Therefore,

$$\lambda_{min}(\Sigma) \frac{\sum_{i=1}^{p-n} \mathcal{X}_i^2(1)}{p - n} \leq \frac{\tilde{y}^T \tilde{y}}{p - n} \leq \lambda_{max}(\Sigma) \frac{\sum_{i=1}^{p-n} \mathcal{X}_i^2(1)}{p - n}.$$

The quantity $\frac{\sum_{i=1}^{p-n} \mathcal{X}_i^2(1)}{p-n}$ can be bounded by Proposition 1. Combining with Proposition 2, we have for any $C > 0$, there exists some $c_3 > 0$ such that

$$P\left( \tilde{y}^T \tilde{y}/(p - n) < c_3 \lambda^{\frac{1}{2}}(\Sigma) \right) \leq e^{-C(p-n)}.$$

Therefore, noticing that $\lambda_{max}^{1/2}(\Sigma)/\lambda_{min}^{1/2}(\Sigma) = \kappa^{1/2}$, $T_1^{(2)}$ can be bounded as

$$P\left( |T_1^{(2)}| > \frac{\sqrt{1 - t_1^2} \kappa^{\frac{1}{2}} t}{\sqrt{c_3}\sqrt{p-n}} \; \middle| \; T_1^{(1)} = t_1 \right) \leq e^{-C(p-n)} + 2e^{-t^2/2}.$$

Using the results from the diagonal term, we have

$$P\left( t_1^2 > c_2 \kappa \frac{n}{p} \right) \leq 2e^{-Cn}. \quad \text{and} \quad P\left( t_1^2 < c_1 \kappa^{-1} \frac{n}{p} \right) \leq 2e^{-Cn}.$$

Consequently, we have

$$P\left( |e_1^T HH^T e_2| > c_4 \kappa t \frac{\sqrt{n}}{p} \right) = P\left( |T_1^{(1)} T_1^{(2)}| > c_4 \kappa t \frac{\sqrt{n}}{p} \; \middle| \; T_1^{(1)} = t_1 \right)$$

$$\leq P\left( T_1^{(1)2} > c_2 \kappa \frac{n}{p} \; \middle| \; T_1^{(1)} = t_1 \right) + P\left( |T_1^{(2)}| > \frac{\kappa^{\frac{1}{2}} t \sqrt{1 - c_1 n/p}}{\sqrt{c_3}\sqrt{p-n}} \; \middle| \; T_1^{(1)} = t_1 \right)$$

$$\leq 5e^{-Cn} + 2e^{-t^2/2},$$

where $c_4 = \frac{\sqrt{c_2(c_0 - 1)}}{\sqrt{c_3(c_0 - 1)}}$. $\qquad \square$

***Proof of Lemma 4.*** Notice that conditioning on $X$, for any fixed index $i$, $e_i^T X^T (XX^T)^{-1} \epsilon$ follows a normal distribution with mean zero and variance $\sigma^2 \| e_i^T X^T (XX^T)^{-1} \|_2^2$. We can first bound the variance and then apply the normal tail bound (8) again to obtain an upper bound for the error term.

The variance term follows

$$\sigma^2 e_i^T X^T (XX^T)^{-2} X e_i \leq \sigma^2 \lambda_{max}\big((XX^T)^{-1}\big) e_i^T HH^T e_i.$$

The $e_i^T HH^T e_i$ part can be bounded according to Lemma 3, while the first part follows

$$\lambda_{max}\big((XX^T)^{-1}\big) = \lambda_{max}\big((Z\Sigma Z^T)^{-1}\big) \leq \lambda_{min}^{-1}(ZZ^T)\lambda_{min}^{-1}(\Sigma) = \frac{\kappa}{p}\lambda_{min}^{-1}(p^{-1}ZZ^T).$$

Thus, using Lemma 4 and 3, we have

$$\sigma^2 \| e_i^T X^T (XX^T)^{-1} \|_2^2 \leq \frac{4\sigma^2 c_2}{(1-c_0^{-1})^2} \frac{n\kappa^2}{p^2}, \tag{9}$$

with probability at least $1 - 4\exp(-Cn)$ if $n > 8C/(c_0-1)^2$. Now combining (9) and (8) we have for any $t > 0$,

$$P\left( |e_i^T X^T (XX^T)^{-1} \epsilon| \geq \frac{2\sigma\sqrt{c_2}\kappa t}{1-c_0^{-1}} \frac{\sqrt{n}}{p} \right) < 4e^{-Cn} + 2e^{-t^2/2}.$$

$\square$

***Proof of Theorem 6.*** The proof depends on Lemma 3 and 4, and a careful choice of the value of $t$ in these two lemmas. We first take union bounds of the two lemmas to obtain

$$P(\min_{i \in Q} |\Phi_{ii}| < c_1 \kappa^{-1} \frac{n}{p}) \leq 2pe^{-Cn},$$

$$P(\max_{i \neq j} |\Phi_{ij}| > c_4 \kappa t \frac{\sqrt{n}}{p}) \leq 5(p^2-p)e^{-Cn} + 2(p^2-p)e^{-t^2/2},$$

and

$$P\left( \|X^T (XX^T)^{-1} \epsilon\|_\infty \geq \frac{2\sigma\sqrt{c_2}\kappa t}{1-c_0^{-1}} \frac{\sqrt{n}}{p} \right) < 4pe^{-Cn} + 2pe^{-t^2/2}.$$

Notice that once we have

$$\min_i |\Phi_{ii}| > 2s\rho \max_{ij} |\Phi_{ij}| + 2\tau^{-1} \|X^T (XX^T)^{-1} \epsilon\|_\infty, \tag{10}$$

then the proof is complete because $\Phi - 2\tau^{-1}\|X^T (XX^T)^{-1}\epsilon\|_\infty$ is already a restricted diagonally dominant matrix. Let $t = \sqrt{Cn}/\nu$. The above equation then requires

$$c_1\kappa^{-1}\frac{n}{p} - \frac{2c_4\sqrt{C}\kappa s\rho}{\nu}\frac{n}{p} - \frac{2\sigma\sqrt{c_2 C}\kappa t}{(1-c_0^{-1})\tau\nu}\frac{n}{p}$$

$$= (c_1\kappa^{-1} - \frac{2c_4\sqrt{C}\kappa s\rho}{\nu} - \frac{2\sigma\sqrt{c_2 C}\kappa}{(1-c_0^{-1})\tau\nu})\frac{n}{p} > 0,$$

which implies that

$$\nu > \frac{2c_4\sqrt{C}\kappa^2\rho s}{c_1} + \frac{2\sigma\sqrt{c_2 C}\kappa^2}{c_1(1-c_0^{-1})\tau} = C_1\kappa^2\rho s + C_2\kappa^2\tau^{-1}\sigma > 1, \tag{11}$$

where $C_1 = \frac{2c_4\sqrt{C}}{c_1}$, $C_2 = \frac{2\sqrt{c_2 C}}{c_1(1-c_0^{-1})}$. Therefore, the probability that (10) does not hold is

$$P\left( \{ (10) \text{ does not hold}\} \right) < (p + 5p^2)e^{-Cn} + 2p^2 e^{-Cn/\nu} < (7 + \frac{1}{n})p^2 e^{-Cn/\nu^2},$$

where the second inequality is due to the fact that $p > n$ and $\nu > 1$. Now for any $\delta > 0$, (10) holds with probability at least $1 - \delta$ requires that

$$n \geq \frac{\nu^2}{C}\bigg( \log(7 + 1/n) + 2\log p - \log \delta \bigg),$$

which is certainly satisfied if (notice that $\sqrt{8} < 3$),

$$n \geq \frac{2\nu^2}{C} \log \frac{3p}{\delta}.$$

Now pushing $\nu$ to the limit as shown in (11) gives the precise condition we need, i.e.

$$n > 2C'\kappa^4(\rho s + \tau^{-1}\sigma)^2 \log \frac{3p}{\delta},$$

where $C' = \max\{\frac{4c_4^2}{c_1^2}, \frac{4c_2}{c_1^2(1-c_0^{-1})^2}\}$. $\qquad\square$