[Reviews · NeurIPS 2015]

Submitted by Assigned_Reviewer_1

The central problem of variable selection is to design a good method for determining the true non-zero coefficients in a linear model. Many methods have been proposed, from forward stepwise to lasso to marginal screening. Marginal screening can be generalized; instead of taking the largest coefficients from X^T y, one can take the largest coefficients from any linear estimator Ay. The idea of HOLP in Wang and Leng (2015) is to choose A to be the pseudoinverse of X. This paper extends that paper by developing a necessary and sufficient condition for screening based on a linear estimator Ay to be "strong screening consistent". That condition is that AX be restricted diagonally dominant (RDD).

This paper is well-written. Since I was only asked to referee this paper lightly, I have not checked the proofs, but if correct, this is an important contribution.
Summary: This paper extends the HOLP method proposed in Wang and Leng (2015) by providing conditions under which it works. That paper was an important contribution to the variable selection literature, and this paper is a nice extension of that result.

Submitted by Assigned_Reviewer_2

The authors propose a two-stage feature selection approach for linear regression. Candidate features are selected in the first stage and the selection is further refined in the second stage. They define "Strong screening consistency" as a criterion for performing the initial stage, and then investigate the implications of that definition when the screening values are computed by a linear operator.

From the content and the references it appears that the authors address the statistics community. My expertise is more in machine learning/computer science, and my review is written from that perspective.

My main point is the following:

The vector beta to be estimated has 0 value for its ith coordinate

iff the ith feature should not be part of the optimal selection. Therefore, identifying the 0 coordinates amounts to finding the optimal selection. My understanding of Definition 2.1 is that

it implies the existence of a threshold on the absolute values of beta_hat coordinates. All and only the zeros of beta are below that threshold. But this means that by trying all p-1 thresholds one can find the optimal solution. Since this problem is known to be NP-hard it is very unlikely

that such strong screening can be computed efficiently.

Additional comments:

The abstract uses many terms that are defined only much later in the paper. These include consistency, rho, sigma, thao, etc.

The idea of (not necessarily consistent) variable screening appears to be, in principle, the same as filters (e.g., Guyon03). It may be a good idea to mention this and give references, or explain the differences.

I am not sure that the O(n^2p) complexity (line 60) is acceptable for variable screening. Naive implementations of forward selection take O(n^2p), and fast implementations take O(np). In this sense O(n^2p) is very slow.

--------------------------------------------------------------------------------------------

Thanks for the rebuttal. It clarified most of the issues I had. I still find strong consistency to be too strong, as it correctly identifies the features to be selected. As explained in the rebuttal, this is an easy case, and may not help the challenge posed by feature selection. Still, I can now appreciate the result much better.

Here is a reference to O(np) forward selection algorithm: Maung et,al. TKDE 2015.

Summary: The paper's definition of Strong Screening Consistency appears to be too strong. Simple complexity arguments seem to suggest that it cannot be calculated efficiently.

Submitted by Assigned_Reviewer_3

This paper presents an unified view of two variable screening methods, SIS and HOLP. More generally, the theory applies to any linear estimators of the form beta = AY for any A. The authors introduced a notion named restricted diagonally dominant (RDD). They also discussed how the RDD condition relates to known conditions in variable selection such as irrepresentable condition for lasso.

I think the paper is well-written and the technical detail is clear. The major contribution is an introduction of the unified theory on SIS and HOLP by using the RDD condition. However, the theory only re-explains the screening consistency guarantees that are already proved for SIS and HOLP in each paper. I think a theory that only explains known results is not strong enough to be accepted. The contribution would be stronger if one can find an insight from the theory how good screening methods can be designed. In particular, it would be interesting if one can construct algorithms better than SIS and HOLP based on the theory.
Summary: This paper presents an unified view of two existing variables screening methods by introducing a notion called restricted diagonally dominant (RDD). Although the theory itself is new, it only re-explains known results and thus I think it is not strong enough to be accepted.

Submitted by Assigned_Reviewer_4

The paper addresses the theoretical properties of linear variable screening methods for the high dimensional setting, p >> n. The authors define the strong screening consistency, which requires the estimator to preserve consistent ordering of the zero and non-zero coefficients and to be sign consistent only on the true support. The main theorem establishes a necessary and sufficient condition for strong screening consistency in the form of the restricted diagonally dominant matrix. This condition is related to the irrepresentable condition (IC) and the restricted eigenvalue condition (REC). The advantage of the proposed condition is that it must hold for a matrix which can be different from the sample covariance matrix, thus leading to strong screening consistency even when the covariates are highly correlated. It is in this setting that Sure Independent Screening (SIS) suffers. Conversely, the high-dimensional ordinary least-square projection (HOLP) can be shown to be strong screening consistent under much weaker conditions.

Overall the paper is well written and structured, presenting the main ideas in a clear manner and relating them to the existing literature. The analysis looks sound even though I did not check all the steps in the proofs. Even though this is a theoretical paper, its motivation mostly comes from applications, where efficacy and efficiency are required. With the ever constant progress in convex optimization algorithms for large scale problems, it would have been interesting to empirically test how much faster linear variable screening is compared to one run of LASSO, for example. Furthermore, in applications, we seldom have a precise knowledge of the parameters rho, tau and s. Can the proposed framework account for model mismatches?
Summary: A neat theoretical paper which addresses consistency of high-dimensional linear variable screening. The proposed framework could be complemented by simple, yet important numerical experiments and a discussion of model-mismatch behaviour.

Submitted by Assigned_Reviewer_5

This paper considers the problem of consistency of high-dimensional screening procedures in linear models. The authors introduce a meaningful notion of strong consistency for screening estimates and establish necessary and sufficient condition for this type of consistency to hold. The authors provide thorough comparison with existing assumptions and show that their condition is often less stringent.

The framework developed in this paper helps provide insight into the properties of various screening procedures including SIS and HOLP. The authors also draw some connections with the existing restricted eigenvalue and irrepresentable conditions. These are interesting contributions.

The paper is well-written and the theoretical analysis is clearly presented.

The notion of strong screening consistency and its connection with restricted diagonal dominance are novel and potentially useful for further theoretical investigations.

I have one question on the role of signal diversity in the sample size requirement for RDD to hold. It is not intuitively clear why screening will suffer if some of the true signal coefficients are high. In other words, should not

\rho depend only on the min signal strength on the support rather than the ratio of the maximum to minimum? This seems like an unnecessary artifact of the analysis, limiting the scope of the theory.
Summary: This paper addresses an interesting question and provides an in-depth, unified analysis that sheds important insight. It is well-written, makes original contribution and I would recommend it for acceptance.

Author Feedback
Author rebuttal: Thanks for all the comments and suggestions. Following are our detailed responses.

*Re: Reviewer_1

It is generally true that the feature selection problem is NP-complete. However, such NP-completeness is only meaningful for the worst case, while very often real data sets are far from the worst case. This is the fundamental motivation for most polynomial runtime feature selection algorithms, including the algorithms in this article. The results of this paper can be summarized in two simple points. First, we found a rough boundary (RDD) condition that distinguish all the data sets between "easy case" and "hard case", while for "easy case" data sets, a simple algorithm such as linear screening is able to solve the model selection problem. Second, we prove that when the regressors X are drawn from some common distributions, with high probability, this data set will be an "easy case".

Regarding the computation complexity, a full forward selection algorithm possesses an O(n^4p) complexity if n variables need to be selected. (Assume we have d variables at the current step, to select a new variable, one needs to fit a regression with d + 1 variables for all p - d remaining variables, which takes O(dn^2p) time, and the total would be O(d^2n^2p). Taking d = n gives the result). We are not sure how the "fast implementation" is done, but even if storing all previous sample covariance matrices, it would still need O(np) time to compute all marginal correlations for each iteration, so the fastest forward selection should have at least O(dnp) complexity, which is equal to O(n^2 p) when d = n.

The O(n^2p) for the linear screening algorithms might look daunting at first glance, but it actually performs better than a lot of O(np) feature selection algorithms. First, the screening algorithm only involves matrix operation without any iterative steps. Therefore, the algorithm can be easily parallelized to have a much lower complexity. Second, the constant before n^2p is small, while many iterative feature selection algorithms that have complexity O(np) actually contain an implicit constant o(k) that depends on the final convergence criterion, which might be large in many situations.

*Re: Reviewer_2

The article provides a common framework to simultaneously study SIS and HOLP, extracting the condition RDD that is needed for both algorithms. The purpose of this framework is exactly for constructing a better screening algorithm than SIS and HOLP. As is shown in the article, RDD is independent of the true value of beta. This means that the condition is fully verifiable once X and A are given. X is a KNOWN matrix, so it is even possible for someone to numerically construct the ancillary matrix A such that AX is close to an identity matrix without using any formula.

In fact, HOLP attempts to construct the ancillary matrix A such that AX = I. It pursues this goal by minimizing the Frobenius norm between the two

min |AX - I|_F

It is interesting to construct the ancillary matrix by solving AX = I under different norms and whether that would achieve better performance than both SIS and HOLP.

*Re: Reviewer_3

In Wang and Leng (2015), the authors provided many numerical experiments for comparing screening methods with conventional variable selection methods such as lasso and scad.

*Re: Reviewer_4

The signal diversity constraint comes from the relativity nature of linear screening. Notice that the estimator of each beta_i is just a linear combination of all non-zero beta_i's, and the coefficient of the linear combination is some column of the screening matrix Phi. If Phi is exactly identity matrix, then there will be no issue for beta_i's being arbitrarily large or small. However, because Phi is only diagonally dominant and the off-diagonal terms are not exactly zero, if the largest signal is significantly greater than the smallest one --- their ratio is larger than the ratio between the diagonal terms and the off-diagonal terms of Phi, then the estimators of the small signal might be dragged down by the large signal and the screening consistency is ruined. This diversity constraint appears in most previous screening literature. For example, in Assumption 2 of SIS (Lv and Fan, 2008), var(Y) is assumed to be bounded, which is roughly a bound on some norm of beta.

*Re: Reviewer_5

For empirical performance, we refer readers to Wang and Leng (2015), which provided many numerical examples comparing different approaches. Because most screening algorithms are based on random designs, the assumptions are mainly distributional assumptions on the predictors X, especially the covariance matrix of X. SIS requires the covariance to satisfy RDD, which is true for most sparse covariance matrices. HOLP and forward regression need a constraint on the conditional number of the covariance, which is actually satisfied by more covariance classes (such as latent factor structure and power-decaying structure).